# Genome Sequence of the Plant-Growth-Promoting Endophyte *Curtobacterium flaccumfaciens* Strain W004

Vladimir K. Chebotar [1,*], Maria S. Gancheva [1,2], Elena P. Chizhevskaya [1], Maria E. Baganova [1], Oksana V. Keleinikova [1], Kharon A. Husainov [3] and Veronika N. Pishchik [1]

[1] All-Russia Research Institute for Agricultural Microbiology, 196608 Saint Petersburg, Russia; m.gancheva@spbu.ru (M.S.G.); chizhevskaya@yandex.ru (E.P.C.); mashul991@mail.ru (M.E.B.); ksu.sha09@yandex.ru (O.V.K.); veronika-bio@rambler.ru (V.N.P.)

[2] Department of Genetics and Biotechnology, Faculty of Biology, Saint Petersburg State University, 199034 Saint Petersburg, Russia

[3] Chechen Research Institute of Agriculture, 366021 Grozny, Chechen Republic, Russia; haron-h14@mail.ru

[*] Correspondence: vladchebotar@rambler.ru or vladchebotar@arriam.ru

**Abstract:** We report the whole-genome sequences of the endophyte *Curtobacterium flaccumfaciens* strain W004 isolated from the seeds of winter wheat, cv. Bezostaya 100. The genome was obtained using Oxford Nanopore MinION sequencing. The bacterium has a circular chromosome consisting of 3.63 kbp with a G+C% content of 70.89%. We found that *Curtobacterium flaccumfaciens* strain W004 could promote the growth of spring wheat plants, resulting in an increase in grain yield of 54.3%. Sequencing the genome of this new strain can provide insights into its potential role in plant–microbe interactions.

**Dataset:** The raw sequencing data and assembled genome are available from the National Center for Biotechnology Information database under BioProject accession number PRJNA1003260.

**Keywords:** *Curtobacterium flaccumfaciens*; draft genome; Nanopore sequencing; plant-growth promotion

## 1. Summary

*Curtobacterium flaccumfaciens* is a Gram-positive bacterium belonging to the family Microbacteriaceae. *C. flaccumfaciens* is known to cause plant diseases such as bacterial wilt, leaf spot, and blight in a variety of crops [1–3]. It has also been reported to be a causative agent of human infections [4,5]. In addition to its pathogenic properties, *C. flaccumfaciens* has been found to have beneficial effects on plants, such as promoting growth and enhancing stress tolerance [6,7]. This has led to research on the potential use of this bacterium as a biofertilizer and biocontrol agent in agriculture. Overall, *C. flaccumfaciens* is an important bacterium with both harmful and beneficial effects. Further research is needed to fully understand its role in plant and human health, as well as its potential for agricultural applications.

## 2. Data Description

Here, we report the whole-genome sequences of the endophyte *Curtobacterium flaccumfaciens* strain W004 isolated from the seeds of winter wheat (*Triticum aestivum* L.), cv. Bezostaya 100, grown in chernozem soil in a mountain and forest zone of the Vedeno region, Chechen Republic, Russia. Nanopore sequencing produced 18,044 reads with a read N50 of 25.1 kb and genome coverage of ~34× The bacterium has a circular chromosome consisting of 3,631,657 bp with a G+C% content of 70.89%. The assembly was circularized and additionally annotated upon submission to the NCBI database. The NCBI Prokaryotic

Genome Annotation Pipeline [8] identified 1 202 pseudogenes that may have been caused by indels, which are common in assemblies using only Nanopore sequencing.

Genome analysis showed that the strain W004 had high similarity with GCM10011263 *Curtobacterium flaccumfaciens*. In the genome of *C. flaccumfaciens* strain W004 were detected clusters for the biosynthesis of antibiotics (merochlorins, microansamycin, arginomycin, bottromycin A2), farnesyl transferase inhibitors (pepticinnamin E), and siderophores (desferrioxamine E) (Table 1).

**Table 1.** Biosynthetic gene clusters predicted in *Curtobacterium flaccumfaciens* W004 by antiSMASH [9].

| Region | Type | From | To | Most Similar Known Cluster | | Similarity |
|---|---|---|---|---|---|---|
| Region 1 | Betalactone | 112,356 | 138,201 | Microansamycin | Polyketide | 7% |
| Region 2 | NAPAA | 794,298 | 828,195 | Arginomycin | Other | 13% |
| Region 3 | NRPS-like T3PKS | 1,399,133 | 1,441,514 | Pepticinnamin E | NRP + polyketide | 12% |
| Region 4 | | 1,625,623 | 1,666,202 | Merochlorin A/merochlorin B/deschloro-merochlorin A/deschloro-merochlorin B/isochloro-merochlorin B/dichloro-merochlorin B/merochlorin D/merochlorin C | Terpene + polyketide: type III polyketide | 9% |
| Region 5 | NI-siderophore | 2,005,682 | 2,017,316 | Desferrioxamine E | Other | 100% |
| Region 6 | Butyrolactone | 2,475,794 | 2,486,471 | | | |
| Region 7 | Terpene | 2,721,328 | 2,742,182 | Carotenoid | Terpene | 28% |
| Region 8 | T3PKS | 3,252,954 | 3,293,358 | Bottromycin A2 | RiPP: bottromycin | 6% |

The effect of the endophytic strain W004 on spring wheat (*Triticum aestivum* L.) cv. Leningradskaya 6 and lettuce (*Lactuca sativa* L.) cv. Great Lakes 659 seeds and plant biomass was studied for two years in greenhouse experiments. It was demonstrated that the strain W004 significantly increased plant biomass by 31.4% and grain yield by 54.3% in spring wheat compared to the non-treated plants (Figure 1a). At the same time, there was no effect of W004 treatment on *L. sativa* biomass ($p$-value = 0.3338) (Figure 1b).

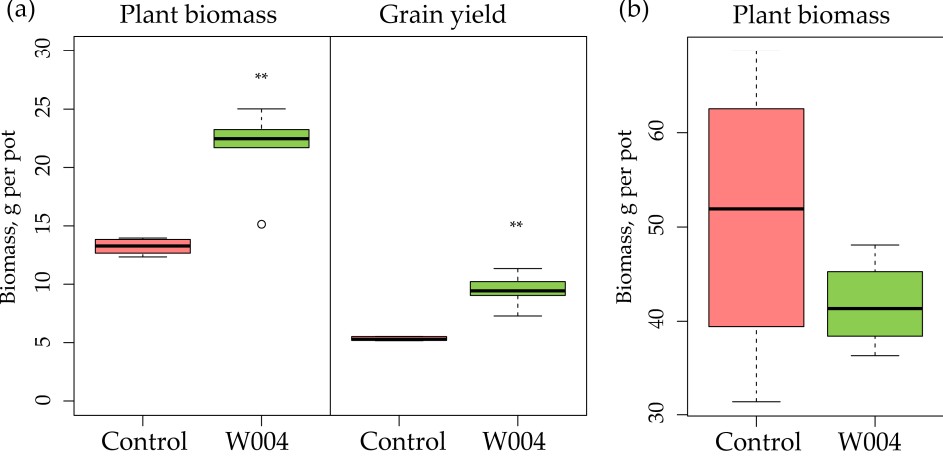

**Figure 1.** The effect of the endophyte *C. flaccumfaciens* W004 on the seed and plant biomass of spring wheat *Triticum aestivum* (**a**) and lettuce *Lactuca sativa* (**b**) compared with control. ** $p$-value < 0.01.

In summary, our analysis of the genome of *C. flaccumfaciens* W004 revealed the presence of clusters for the biosynthesis of antibiotics and siderophores. These compounds may play a role in creating a favorable environment that promotes plant growth.

## 3. Methods

### 3.1. Bacteria Isolation and DNA Extraction

Bacteria isolation was performed as previously described [10]. DNA was extracted from a single colony using the cetyltrimethylammonium bromide (CTAB)-NaCl method [11].

### 3.2. Genome Sequencing and Assembly

The whole genome was sequenced in the Core Centrum "Genomic Technologies, Proteomics and Cell Biology" at the All-Russia Research Institute for Agricultural Microbiology (ARRIAM) using a MinION device with a 9.4.1 flow cell (Oxford Nanopore, Cambridge, UK). Raw sequence data were base-called using Guppy v3.3.0 [12] in the high-accuracy mode, and a total of 17,370 raw reads were generated. The reads were assembled by Flye v2.9 [13] and contigs were polished by Racon v. 1.3.2 [14] (4 iterations with parameters -m 8 -x -6 -g -8 -w 500) and Medaka v. 1.4.3 (https://github.com/nanoporetech/medaka ((accessed on 1 July 2023)). The quality of the final assembly was evaluated using QUAST v 5.1.0 [15]. Genome annotation of the assembly was performed using Prokka [16]. The annotation revealed 4,605 coding DNA sequences and 65 RNA sequences in the assembly (55 tRNAs, 9 rRNAs, and 1 tmRNA). Default settings were used for all software unless otherwise noted. gcType [17] was used for genome analysis. The secondary metabolites' biosynthetic gene clusters were identified using AntiSMASH 7.0 [9].

### 3.3. Plant-Growth Promotion

The ability of strain W004 to promote the growth of spring wheat (*Triticum aestivum* L.) cv Leningradskaya 6 and lettuce (*Lactuca sativa* L.) cv Great Lakes 659 plants was studied in two-year experiments under greenhouse conditions in pots with soil (nine and three plants per pot, respectively). Seeds of both plant species were treated with a bacterial suspension containing $10^8$ CFU per mL of strain W004 for 30 min. Box plots were drawn using the R boxplot () function. Student's *t*-test was used to compare the means of the two groups.

**Author Contributions:** Conceptualization, V.K.C. and M.S.G.; methodology, M.S.G. and E.P.C.; software, M.S.G.; validation, M.S.G., E.P.C., and V.N.P.; formal analysis, M.S.G. and E.P.C.; investigation, E.P.C., M.E.B., and O.V.K.; resources, K.A.H. and V.N.P.; data curation, V.N.P.; writing—original draft preparation, M.S.G.; writing—review and editing, M.S.G. and V.K.C.; visualization, M.S.G.; supervision, V.K.C.; project administration, V.K.C.; funding acquisition, M.S.G. and V.K.C. All authors have read and agreed to the published version of the manuscript.

**Funding:** This work was funded by the Ministry of Science and Higher Education of the Russian Federation in accordance with agreement no. 075-15-2021-1055, 28 September 2021, on providing a grant in the form of subsidies from the federal budget of the Russian Federation. The grant was provided for the implementation of the project Mobilization of the Genetic Resources of Microorganisms on the Basis of the Russian Collection of Agricultural Microorganisms (RCAM) at the All-Russia Research Institute for Agricultural Microbiology (ARRIAM) according to the Network Principle of Organization.

**Institutional Review Board Statement:** Not applicable.

**Informed Consent Statement:** Not applicable.

**Data Availability Statement:** The genome sequence was deposited in GenBank (accession number CP133386). The raw reads were deposited in the NCBI Sequence Read Archive under accession number SRR25592758.

**Acknowledgments:** This research was performed using equipment of the Core Centrum "Genomic Technologies, Proteomics and Cell Biology" in ARRIAM.

**Conflicts of Interest:** The authors declare no conflict of interest.

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
