# Peer review of "Genome Sequence of the Plant-Growth-Promoting Endophyte Curtobacterium flaccumfaciens Strain W004"

_data_

Round 1
Reviewer 1 Report
Comments and Suggestions for Authors
Summary:
The article reports the whole-genome sequences of the endophyte Curtobacterium flaccumfaciens strain W004, isolated from winter wheat seeds (cv.Bezostaya 100). The genome, obtained through Oxford Nanopore MinION sequencing, comprises a circular chromosome of 3,63 kbp with a G+C% content of 70.89%. Notably, this bacterium, identified as Curtobacterium flaccumfaciens strain W004, demonstrates the ability to enhance the growth of spring wheat plants, resulting in a remarkable 54.3% increase in grain yield. The sequencing of this strain's genome offers valuable insights into its potential role in plant-microbe interactions.
The authors identified several interesting genome regions presented in Table 1. A comprehensive summary of the method including the genome assembly is included.
I find the study appropriate for publishing in its present form; however, I have one minor suggestion.
Suggestion: I would recommend including a direct URL link to the sequencing data deposited at the National Center for Biotechnology (BioProject PRJNA1003260). This link will provide convenient easy access to detailed genomic information for further exploration and analysis.
Comments on the Quality of English LanguageEnglish style, grammar, and spelling are appropriate.
Reviewer 2 Report
Comments and Suggestions for Authors
The authors report a draft genomic sequence and growth promoting effect of Curtobacterium flaccumfaciens Strain W004 based on HAC called ONP sequences. The use of HAC rather than a SUP model causes numerous INDEL errors as stated in the manuscript.
I am missing information on the ONP data:
* Total number of nucleotides
* N50 of the ONP reads; not the final assembly
* Coverage of the genome
This information indicate the quality of the large-scale assembly of the genome.
Further I notice that no plasmids were found in this strain despite that plasmids are frequently found in Curtobacterium flaccumfaciens strains.
Could the authors state which measures were taken to exclude the presence of plasmid(s) in strain W004?
